# Scikick: A sidekick for workflow clarity and reproducibility during extensive data analysis

**Matthew Carlucci[1,2], Tadas Bareikis[1,2], Karolis Koncevičius[2], Povilas Gibas[1,2], Algimantas Kriščiūnas[2], Art Petronis[1,2], Gabriel Oh[1,2,3]***

**1** The Krembil Family Epigenetics Laboratory, The Campbell Family Mental Health Research Institute, Centre for Addiction and Mental Health, Toronto, Ontario, Canada, **2** Institute of Biotechnology, Life Sciences Center, Vilnius University, Vilnius, Lithuania, **3** Stanford University School of Medicine, Stanford, California, United States of America

* gabriel.oh@stanford.edu

## Abstract

Reproducibility is crucial for scientific progress, yet a clear research data analysis workflow is challenging to implement and maintain. As a result, a record of computational steps performed on the data to arrive at the key research findings is often missing. We developed Scikick, a tool that eases the configuration, execution, and presentation of scientific computational analyses. Scikick allows for workflow configurations with notebooks as the units of execution, defines a standard structure for the project, automatically tracks the defined interdependencies between the data analysis steps, and implements methods to compile all research results into a cohesive final report. Utilities provided by Scikick help turn the complicated management of transparent data analysis workflows into a standardized and feasible practice. Scikick version 0.2.1 code and documentation is available as supplementary material. The Scikick software is available on GitHub (https://github.com/matthewcarlucci/scikick) and is distributed with PyPi (https://pypi.org/project/scikick/) under a GPL-3 license.

## 1. Introduction

Research reproducibility, in the many forms it takes [1–3], is essential to the scientific method. Multiple insights can often be gained from a single large dataset, however, the breadth of such investigations has placed a heavy burden on researchers who aim to practice full computational transparency. It is essential that analytical procedures are clearly documented throughout an investigation, including details of their intent, background rationale, implementation, and analysis outputs [4, 5]. In its absence, investigative decisions, assumptions, and results can lose their context, and in turn, lower the quality of research communication [6].

Computational notebook formats (e.g., Jupyter Notebooks [7] and Rmarkdown [8]) and their associated development environments have paved a path for streamlined generation and sharing of computational results. Notebooks enable investigators to compile the analytical context (i.e., text), implementation (code), and results (figures) within a single document. This results in a report that reflects the entire process of analysis and serves as a transparent lens into how computations unfolded.

**Funding:** This project was supported by the European Social Fund, ec.europa.eu/esf (project No 09.3.3-LMT-K-712-17-0008) under grant agreement with the Research Council of Lithuania (LMTLT; lmt.lt) awarded to A.P. The funders had no role in study design, data collection and analysis, decision to publish, or preparation of the manuscript.

**Competing interests:** The authors have declared that no competing interests exist.

In order to develop larger projects, there is a demand to use multiple notebooks within the same analysis. To this end, in addition to best practice guidelines [9] and improvements to the notebook format [10], tools have been designed for some specific project types; generating reading materials on computational topics (e.g., bookdown [11] and Jupyter Book [12]) or developing software packages fully within notebooks (e.g., nbdev [13]). However, these solutions do not emphasize the ordered and interdependent execution of notebooks common to computational research projects. As such, reproducibility is compromised when projects are not configured to execute notebooks in the correct order, and transparency is compromised when projects do not clearly document this execution order.

The clarity of notebook outputs gives it an advantage to tools whose main purpose is to configure ordered computations (e.g., GNU Make [14], Snakemake [15], Nextflow [16], etc.). To benefit from both toolsets, researchers often use them in tandem to reproducibly configure the execution of a notebook collection. Further, researchers can produce graphical representations of this configuration to transparently represent the execution to the reader. However, assembling and maintaining these configurations throughout evolving projects is cumbersome. Therefore, many rapidly developing projects cannot dedicate the resources necessary for this level of transparency and reproducibility (Fig 1a).

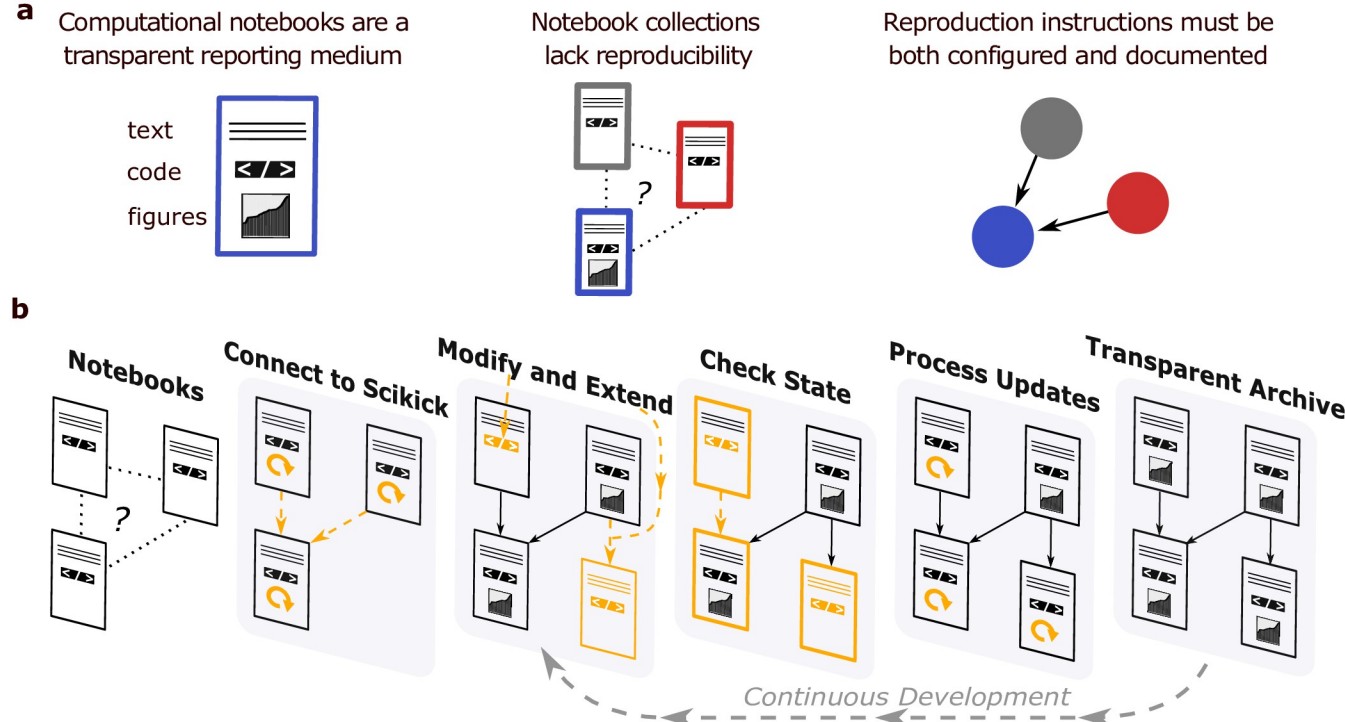

**Fig 1. Scikick workflow development use-case, practices, and features. a)** An illustration of the problem Scikick aims to address. Left) A schematic of a rendered computational notebook with contextual descriptions accompanying code and results demonstrating the clarity of the notebook format. Centre) A minimal "notebook collection" where execution order of notebooks is undocumented and not configured, compromising both transparency and reproducibility. Right) A graphical representation of a workflow management configuration which supplements the notebook collection to execute the notebooks in the specified order. **b)** The illustration shows the main Scikick features used to manage a collection of notebooks throughout a project. An unstructured collection of notebooks are initially executed by Scikick to generate a structured report. New content inside the workflow, including modifications to a notebook, upstream modifications or the addition of new notebooks, are all detected by Scikick. Pending updates are computed by Scikick to generate an up-to-date report which can be easily navigated. Users repeatedly apply the modification, state management, and execution features throughout a project's development.

To simplify and improve transparency during these projects, we developed Scikick, a minimalistic command-line utility for maintaining computational notebook workflows. Scikick integrates the notebook format, workflow tools, and other elements necessary to streamline data-intensive research [17]. Our tool is designed to promote the maintenance of computational workflow archives that are easily inspectable by a diverse scientific readership at all project stages, conceptually extending the benefits of computational notebooks to computational workflows.

## 2. Results

### 2.1 Feature overview

The features of Scikick are designed to maintain structure and re-executability while performing an investigation which spans multiple computational notebooks that may depend on one-another. Notebooks are arranged into a workflow graph using Scikick's command-line interface, implemented with Python ver. 3.6+, to update the configuration file and inspect the workflow state (Fig 1b). Notebooks are executed through Scikick's Snakemake workflow configuration and are compiled into a static report containing all data analysis outputs. Scikick features are broadly separated into notebook, workflow, and repository management functionalities that seamlessly work together.

**2.1.1 Command-line interface overview.** The main commands for accessing this functionality are: "init", "add", "run", and "status". The "sk init" command checks for software requirements and creates an empty configuration file within the project directory. The "sk add" command is used to add notebooks and define the workflow graph. "sk run" calls on Snakemake to execute notebooks in the specified order. Then, as notebooks are modified, the "sk status" command displays which notebooks require (re)execution. Further usage information can be found within the Scikick documentation and command-line help outputs.

### 2.2 Notebook execution and metadata capture

To promote easy access for recording results in a notebook format, Scikick supports a plurality of file and notebook formats that are automatically executed using appropriate methods (e.g., R scripts are converted to Rmarkdown) to capture code, console, and graphical outputs in a markdown document which would otherwise go unrecorded. All notebook outputs are compiled with Rmarkdown into a cohesive final report where the report navigation, which is automatically configured by Scikick, allows readers to easily identify the notebook source file. Scikick also logs useful technical information related to each notebook into the resulting markdown files, such as software versions, user defined functions, execution time, and versioning history of the notebook. Usage of these features increases the amount of information available to readers revisiting old computational archives.

### 2.3 Concurrent management of automation and reporting

As a project matures and more analyses are added, it is reasonable to expect that long notebooks will be broken up into many smaller notebooks [9] in order to focus on specific topics, reduce resource usage (e.g., memory and runtime), and simplify computing environment namespaces. Creation of notebooks which depend on one-another during this stage puts new strains on automating the workflow while also ensuring the report remains transparent. Therefore, Scikick provides simple features to define dependence between notebooks and to manage each notebook's state. After authoring a new notebook and placing it in the workflow graph, the status of all notebooks can be verified by the user. Any pending notebook executions will

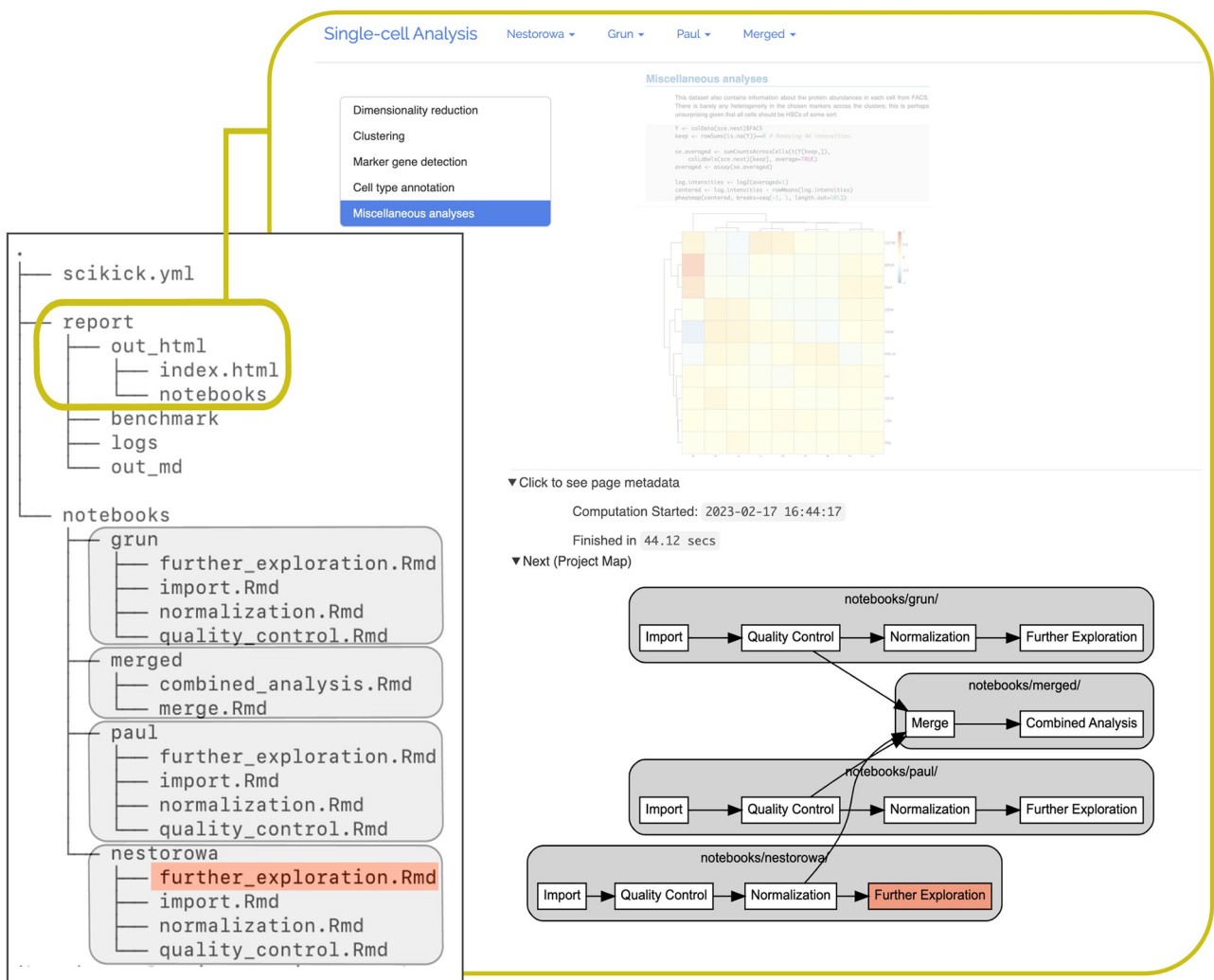

**Fig 2. Report and project map generated by Scikick for the single-cell RNA sequencing analysis demonstration.** A series of notebooks shown in the file tree under the "notebooks" directory were added to the project's "scikick.yml" YAML configuration file using the command-line interface. The notebooks were executed to generate the displayed HTML archive (orange border) which resides in the "report" directory. A workflow graph ("Project Map") was automatically generated by Scikick using the graphviz.dot format [18]. The graph illustrates the order of notebook executions, outlines the organization of the project's source code, and allows for report navigation such that a reader can trace the provenance of results among the notebooks. Each grouping (grey box) represents a series of notebooks that were used to analyze an individual transcriptomic dataset with a final group containing two notebooks used to perform a combined analysis of the datasets. The red node represents the current notebook results page being viewed in the report. A table of contents is provided on the left for each page in the report.

run in an appropriate order to overwrite older outputs such that results represent a procedural execution of each notebook's code. Finally, embedding notebook results and the order of notebook executions into a final report in a human-readable format allows for the report archive to be fully understood by readers (e.g., Fig 2).

### 2.4 Demonstrations and guides

A series of vignettes in the Scikick documentation (S1 File; https://github.com/matthewcarlucci/scikick) includes a quick-start guide, a comprehensive user manual, details

on methods and design, as well as a demonstration of Scikick's capabilities through an analysis of single-cell RNA sequencing (scRNA-seq) datasets [19]. Here, we demonstrated how the implementation of a complex workflow involving data inspection, quality control, normalization, dimensionality reduction, and data exploration, is managed through the usage of Scikick. This allowed the pursuit of nonlinear avenues of investigation across multiple datasets and project stages and automatically embedded a navigable graph of the workflow within the archive (Fig 2).

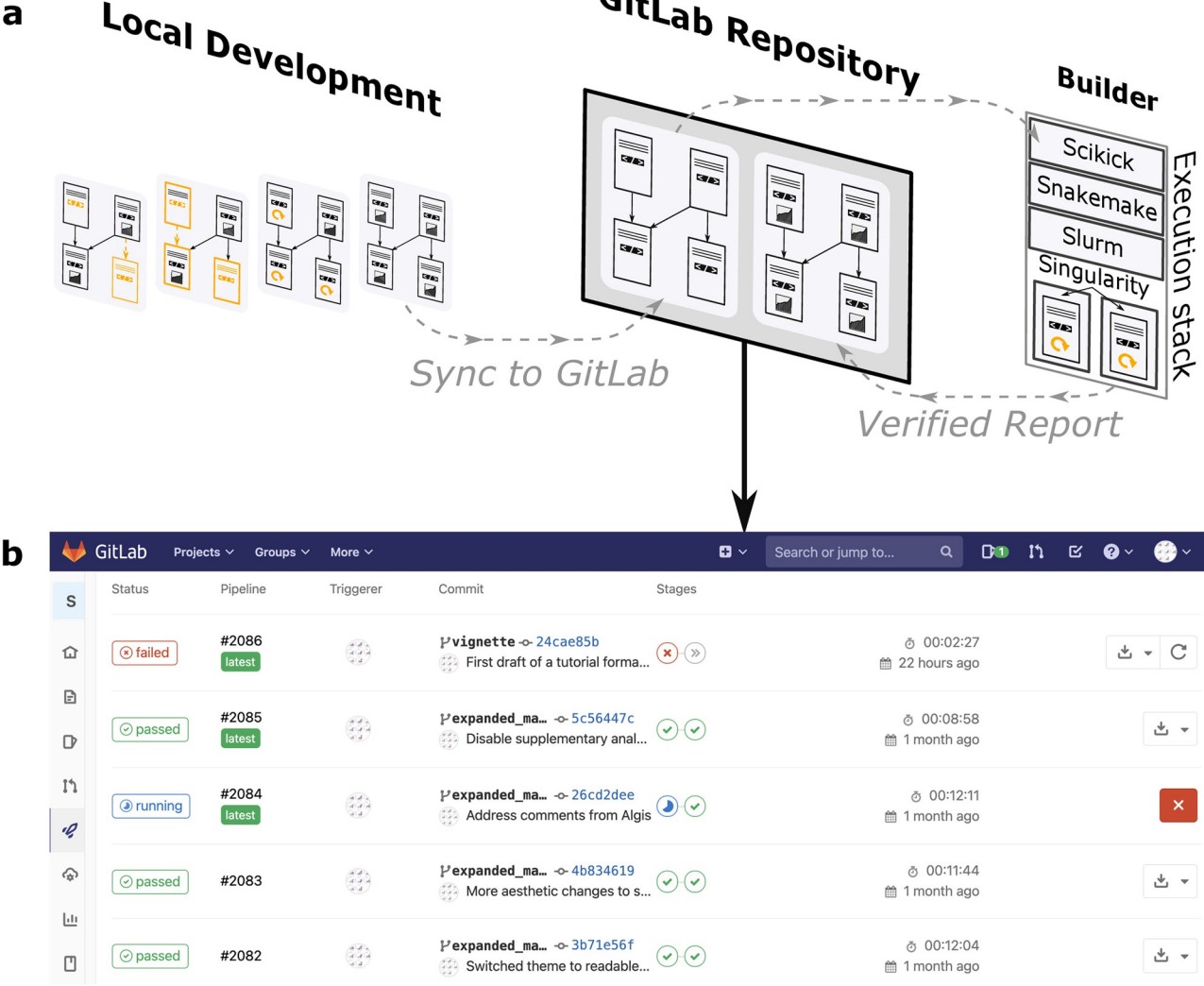

**Fig 3. Developing Scikick projects to maturation with continuous analysis. a)** This schematic illustration represents a configuration for an automated re-execution of a Scikick project. Local development of notebook collections with Scikick proceeds as described in Fig 1b. Upon pushing a version of the project's source code to a remote server (GitLab), the project is executed utilizing resources of a SLURM compute cluster where each notebook is submitted in parallel as cluster jobs. To provide fixed software environments for the remote machines, Singularity containers are defined in the Scikick configuration file. The final reproducible report is stored alongside the code that produced it as a "verified report", which is known to have executed from start to finish within the specified software environment. **b**) A sample of the 'Continuous Integration' section of GitLab shows versions of the analysis that are executed as they are "pushed" to the GitLab server, as configured in (**a**). Some versions encountered errors which were rapidly addressed which may have been detected much later if re-execution was not performed. Successful analysis verification runs store a final Scikick report on the server, alongside the source code that produced it, where results may be inspected at any time in the future, and further, may be precisely reproduced with usage of the specified container.

### 2.5 Project management with robust versioned archives

Advanced practices such as "Continuous Analysis" [20] provide schemes for producing verifiable end-to-end runs of computational analyses by ensuring that code can execute from start to finish within a specified software environment. Despite having verified a machine-readable workflow in this way, maintaining an archive that is also human-readable is a difficult, but necessary, complementary task [9]. Crucial to this practice, Scikick projects remain reader-ready throughout their development with the use of features such as a simplistic configuration and a navigable project graph (Fig 2). Additionally, Scikick projects can be stored entirely as plain text and sets a clear structure for which files are tracked with version control, further simplifying the adoption of Continuous Analysis on version control platforms (e.g., GitHub, GitLab, BitBucket, etc.). Lastly, demanding knowledge needed to use advanced computing is reduced for Scikick projects; minimal additional configuration can achieve distributed notebook executions over a cluster's resources in a prespecified container environment (e.g., Docker [21], Singularity [22], etc.). Since all Scikick projects use the same execution and reporting methods, a single configuration of continuous analysis on a given set of computational infrastructure can be applied to all projects. We implemented such a configuration on common academic infrastructure which monitors source code for changes and re-runs scientific analysis automatically (Fig 3). Altogether, Scikick methods are sufficiently flexible to implement as part of a variety of computing infrastructure and environments.

## 3. Conclusion

Computational notebooks have provided transparency to research reporting, and yet require supporting tooling for the analysis of large-scale datasets with many investigative branches. Projects lack access to streamlined build tools to support computational notebooks, and therefore lose the reproducibility and transparency needed for auditing analyses. Scikick provides the ability to maintain clarity of data analysis and research workflow development all the way through to verified analysis archives. Scikick can be used to execute polyglot projects, and underlying tooling updates (e.g., Quarto [23]) may improve language agnosticism in the future. Well-structured and annotated investigative projects, supported by the use of tools like Scikick, will help improve transparency to allow for more rigorous review and thereby improve reliability of scientific works.

## Supporting information

**S1 File. Scikick 0.2.1 source code and documentation.**
(ZIP)

## Acknowledgments

The authors thank all of those involved in providing comments throughout the development of Scikick which provided valuable feedback for the design and usage of the tool.

## Author Contributions

**Software:** Matthew Carlucci, Tadas Bareikis, Karolis Koncevičius, Povilas Gibas, Algimantas Kriščiūnas.

**Supervision:** Karolis Koncevičius, Art Petronis, Gabriel Oh.

**Writing – original draft:** Matthew Carlucci, Gabriel Oh.

**Writing – review & editing:** Matthew Carlucci, Tadas Bareikis, Karolis Koncevičius, Povilas Gibas, Algimantas Kriščiūnas, Art Petronis, Gabriel Oh.

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
