## [Decision Letter · Decision Letter 0]

9 May 2023

PONE-D-23-11089Scikick: a sidekick for workflow clarity and reproducibility during extensive data analysisPLOS ONE

Dear Dr. Oh,

Thank you for submitting your manuscript to PLOS ONE. After careful consideration, we feel that it has merit but does not fully meet PLOS ONE’s publication criteria as it currently stands. Therefore, we invite you to submit a revised version of the manuscript that addresses the points raised during the review process.

We look forward to receiving your revised manuscript.

Kind regards,

Anna Bernasconi, PhD

Academic Editor

PLOS ONE

Journal Requirements:

Additional Editor Comments (if provided):

Dear authors the paper contributes an important work. Please carefully revise the manuscript addressing the comments of Rev. 1 and 2. Provide a revised manuscript with changes/additions highlighted in different color.

We will be glad to review your revised manuscript once the points have been addressed.

Reviewers' comments:

Reviewer's Responses to Questions

**Comments to the Author**

1. Is the manuscript technically sound, and do the data support the conclusions?

Reviewer #1: Yes

Reviewer #2: Yes

Reviewer #3: Yes

2. Has the statistical analysis been performed appropriately and rigorously? 

Reviewer #1: N/A

Reviewer #2: N/A

Reviewer #3: N/A

3. Have the authors made all data underlying the findings in their manuscript fully available?

Reviewer #1: Yes

Reviewer #2: Yes

Reviewer #3: Yes

4. Is the manuscript presented in an intelligible fashion and written in standard English?

Reviewer #1: Yes

Reviewer #2: Yes

Reviewer #3: Yes

5. Review Comments to the Author

Reviewer #1: In this work, authors present Scikick, an interesting tool to create workflows composed by notebooks rather than from scripts or executable files as we all are used to do.

I have never heard about a similar tool and many times wondered how that could be done, so my overall opinion is that this is a good addition to the bioinformatics software ecosystem.

Authors have made a great effort to automatically run notebooks using Snakemake and provide exhaustive documentation to get started with the tool, including basic and advanced usage examples.

Regarding the paper, I have struggled a little bit with figure 2. I was able to fully understand it after browsing the example report online and navigating trough the different sections, so my suggestion is try improving the figure 2 legend so that readers can understand it easier. For instance, you could add a files tree of the project and briefly explain them.

In addition, three points:

- I'd suggest authors adding a new section (and maybe a figure) to explain the basic commands of the "sk" tool (init, add, status and run) as it is important for readers to see how the tool works and how dependencies are defined.

- Have you considered including tools or some mechanisms to obtain user-friendly readable diffs between notebook files? By default, "git diff" commands on Python notebooks are not very useful.

- Regarding figure 3, what does mean "verified archive"? I think that this figure can be also improved. For me, the flow would go from local development to the GitLab server and from that point the execution of the "continuous analysis" triggered. With the current image, one can think that the continuous analysis is performed in the same computer as the workflow is being developed.

Reviewer #2: The authors describe a command-line software package for tracking the sequence in which Jupyter notebooks are executed and "to define dependence between notebooks and to manage each notebook’s state." The goal is to prevent scientists from having to perform these tasks manually, which can be time consuming and error prone. For scientists who use notebooks, this tool may be helpful. Though in my own personal case, I have not yet found a scenario where I would use this tool.

The writing is clear, the software appears to be well designed, and it is open source. Below are some comments as suggestions for improving the paper.

- Figure 1 seems a bit superfluous to me. It attempts to provide a high-level justification for the tool, but in my opinion, it doesn't add anything to the paper beyond what is already in the text.

- The paper focuses mostly on justifying the need for the tool. But it provides little insight on *how* the tool works. For example, how does it collect the metadata and integrate everything? How does it integrate with snakemake? Etc. I don't think it's necessary to provide a lot more detail. But a little more would be helpful.

- The paper says, "Additionally, maintaining a simple project structure suitable for version control, further eases the adoption of Continuous Analysis on version control platforms (e.g., GitHub, GitLab, BitBucket, etc.)." I'm not sure if I understand exactly what this is saying. How does it support using version control?

- I'm probably missing it, but I don't see any explicit instructions about how to install the software. That's a pretty standard thing to see on documentation sites.

Reviewer #3: Computational analyses are getting a central part in an ever increasing number of scientific experiments and the amount and complexity of code that is produced is often difficult to maintain, distribute and replicate in a standard way.

In this manuscript the authors present a tools for easy integration, maintenance and organisation of multiple computational notebooks linked to complex analysis tasks. The tool functionalities are well structured and provide a way to add a layer of organisation to code without affecting the internal logic of existing components. The manuscript is clear and well written.

I think that, if adopted, the proposed tool can bring a considerable benefit to the research community.

6. PLOS authors have the option to publish the peer review history of their article (what does this mean?). If published, this will include your full peer review and any attached files.

Reviewer #1: **Yes: **Hugo López-Fernández

Reviewer #2: No

Reviewer #3: No

---

## [Author Response · Author response to Decision Letter 0]

21 Jun 2023

Reviewer #1:

Comment: Regarding the paper, I have struggled a little bit with figure 2. I was able to fully understand it after browsing the example report online and navigating through the different sections, so my suggestion is try improving the figure 2 legend so that readers can understand it easier. For instance, you could add a files tree of the project and briefly explain them.

Response: We have taken the reviewers suggestion to better highlight the merits of Scikick being displayed in figure 2 by including the file tree and further describing the figure (L133-L139).

Comment: I'd suggest authors adding a new section (and maybe a figure) to explain the basic commands of the "sk" tool (init, add, status and run) as it is important for readers to see how the tool works and how dependencies are defined.

Response: In our initial submission, we have included detailed information regarding the execution in our supplemental information in an effort to reduce too much technicality in the manuscript. Based on the reviewer’s feedback, we have included a new section as an overview of the command-line interface (L94-L103), yet direct to the documentation for longer form tutorials and examples.

Comment: Have you considered including tools or some mechanisms to obtain user-friendly readable diffs between notebook files? By default, "git diff" commands on Python notebooks are not very useful.

Response: Thank you for the suggestion. While it is an interesting idea, we think it is beyond the scope of Scikick. There are some dedicated tools to specifically address this challenging issue (e.g., https://nbdime.readthedocs.io/en/latest/), and we feel these specialty tools and others like it will work in tandem with Scikick.

Comment: Regarding figure 3, what does mean "verified archive"? I think that this figure can be also improved. For me, the flow would go from local development to the GitLab server and from that point the execution of the "continuous analysis" triggered. With the current image, one can think that the continuous analysis is performed in the same computer as the workflow is being developed.

Response: We have replaced “verified archive” in Figure 3 with “GitLab Repository” in order to avoid the possible misunderstanding pointed out by the reviewer. We also have clarified what we mean by “verified” throughout the revised manuscript (L160-L161; L188-L190). 

Reviewer #2: 

Comment: Figure 1 seems a bit superfluous to me. It attempts to provide a high-level justification for the tool, but in my opinion, it doesn't add anything to the paper beyond what is already in the text.

Response: We agree with the reviewer that it is somewhat repetitive to what is described in the main text. However, we believe that an overview may serve a similar role to graphical abstracts for those who are trying to rapidly process information, as a visual aid.

Comment: The paper focuses mostly on justifying the need for the tool. But it provides little insight on *how* the tool works. For example, how does it collect the metadata and integrate everything? How does it integrate with snakemake? Etc. I don't think it's necessary to provide a lot more detail. But a little more would be helpful.

Response: Much of this information was included as supplemental information (found in the documentation “core design”) or was custom code that would be trivial to explain (e.g. “sys.time” for logging runtime). We used the article to highlight the need for tools like Scikick and the utility they add (i.e., the why rather than the how). We have made additions to the manuscript where appropriate to better allude to this content and to the underlying architecture of the software (L91; L109; L110; L112; L133-L136; L165-L166). 

Comment: The paper says, "Additionally, maintaining a simple project structure suitable for version control, further eases the adoption of Continuous Analysis on version control platforms (e.g., GitHub, GitLab, BitBucket, etc.)." I'm not sure if I understand exactly what this is saying. How does it support using version control?

Response: This sentence combined too many concepts and has now been expanded for clarity (L165-L167). 

Comment: I'm probably missing it, but I don't see any explicit instructions about how to install the software. That's a pretty standard thing to see on documentation sites.

Response: The installation instructions should have been found on the homepage (https://petronislab.camh.ca/pub/scikick). We have changed this link from “Start using Scikick” to “Installation and Usage” to avoid confusion and have added a “Quick-Start” installation to the GitHub homepage to further improve accessibility.

---

## [Decision Letter · Decision Letter 1]

13 Jul 2023

Scikick: a sidekick for workflow clarity and reproducibility during extensive data analysis

PONE-D-23-11089R1

Dear Dr. Oh,

We’re pleased to inform you that your manuscript has been judged scientifically suitable for publication and will be formally accepted for publication once it meets all outstanding technical requirements.

Kind regards,

Anna Bernasconi, PhD

Academic Editor

PLOS ONE

Additional Editor Comments (optional):

Dear authors,

your revised manuscript has been re-assessed by two reviewers and myself. We gladly recommend the paper for acceptance.

Reviewers' comments:

Reviewer's Responses to Questions

**Comments to the Author**

1. If the authors have adequately addressed your comments raised in a previous round of review and you feel that this manuscript is now acceptable for publication, you may indicate that here to bypass the “Comments to the Author” section, enter your conflict of interest statement in the “Confidential to Editor” section, and submit your "Accept" recommendation.

Reviewer #1: All comments have been addressed

Reviewer #2: All comments have been addressed

2. Is the manuscript technically sound, and do the data support the conclusions?

Reviewer #1: Yes

Reviewer #2: Yes

3. Has the statistical analysis been performed appropriately and rigorously? 

Reviewer #1: N/A

Reviewer #2: Yes

4. Have the authors made all data underlying the findings in their manuscript fully available?

Reviewer #1: Yes

Reviewer #2: Yes

5. Is the manuscript presented in an intelligible fashion and written in standard English?

Reviewer #1: Yes

Reviewer #2: Yes

6. Review Comments to the Author

Reviewer #1: All comments have been addressed so the paper can be accepted now. I do not have anything else to add.

Reviewer #2: I have no additional comments.This form is asking me to write a statement that is at least 100 characters long, but I have nothing else to say... :)

7. PLOS authors have the option to publish the peer review history of their article (what does this mean?). If published, this will include your full peer review and any attached files.

Reviewer #1: **Yes: **Hugo López-Fernández

Reviewer #2: **Yes: **Stephen Piccolo

---

## [Editor Report · Acceptance letter]

18 Jul 2023

PONE-D-23-11089R1 

Scikick: a sidekick for workflow clarity and reproducibility during extensive data analysis 

Dear Dr. Oh:

I'm pleased to inform you that your manuscript has been deemed suitable for publication in PLOS ONE. Congratulations! Your manuscript is now with our production department. 

Kind regards, 

on behalf of

Dr. Anna Bernasconi 

Academic Editor

PLOS ONE